# *Patterny*: A Troupe of Decipherment Helpers for Intrinsic Disorder, Low Complexity and Compositional Bias in Proteins

**DOI:** 10.3390/biom15091332

**Published:** 2025-09-18

**Authors:** Paul M. Harrison

**Affiliations:** Department of Biology, McGill University, Montreal, QC H3A 1B1, Canada; paul.harrison@mcgill.ca

**Keywords:** intrinsic disorder, compositional bias, low complexity, annotation, function, software

## Abstract

Intrinsically disordered regions (IDRs) are sometimes considered parts of the ‘dark proteomes’, i.e., protein parts that have been largely under-appreciated, as are the overlapping phenomena of low-complexity or compositionally biased regions (LCRs/CBRs). Experimentalists and computationalists alike are still learning how to decrypt the functionally meaningful features of such regions. Here, I report the creation of the support troupe ***Patterny*** to aid such protein cryptanalysis. The current troupe members are named *Blocky*, *Bandy*, *Moduley*, *Repeaty*, and *Runny*. To discern important features, protein regions are compared to ideal assortments wherein everything is sampled proportionally and dispersed randomly. *Blocky* discerns the segregation of amino-acids by type, and scores them for it. *Bandy* is focused on picking out compositional bands and calculating their evenness. *Moduley* labels the boundaries of optimized compositional modules (‘CModules’) and other possible boundary sets for compositionally biased regions. *Repeaty* concisely summarizes repetitiveness using an information entropy of amino-acid interval diversity. *Runny* enumerates homopeptide content and assesses its significance. Both original whole sequences and CModules from *Moduley*, are fed into the other ***Patterny*** members. ***Patterny*** is applied to some illustrative sample data from yeast proteome and the DISPROT database. It is available at Github, and might aid those aiming to intensify light-shedding and hypothesis generation for protein regions with function encoded in a distributed manner, such as IDRs and LCRs/CBRs more generally.

## 1. Introduction

Intrinsically disordered regions (IDRs) are protein parts that remain unfolded during at least a part of their functioning. Since the early days of their experimental characterization, computational annotation or prediction of IDRs has been a lively topic of bioinformatical research, with programs such as PONDR, Disopred, and IUPred [1,2,3], and more recent developments applying language models (e.g., AIUPred [4], PUNCH2 [5], AlphaFold3 [6], DisoFLAG-IDR [7]).

IDRs have long been associated with lower ‘sequence complexity’, i.e., the sequences are generally simpler, more repetitive and sample residue types un-evenly [8,9]. Originally, the term ‘low-complexity’ as applied to proteins had a strictly algorithmic meaning, referring to sequence tracts that had lower information entropy, as calculated by the algorithm SEG by Wootton & Federhen [10]. ‘Compositional bias’ is a more general term that covers a range from highly biased and repetitive sequences to those that have a milder compositional skew [10,11]. Arguably, sequence complexity per se is less likely to be under selection in protein sequences than say a specific compositional bias for amino acids that has a functional role, and it is also not clear where an imaginary boundary around the concept of ‘low-complexity’ could be placed [11]. Compositional biases are directly linked to functional roles of IDRs [12,13].

However, IDRs and CBRs are not simply compositional entities. They have various types of patterning such as repeat structure, alternating compositional blocks or bands, multiple discrete compositional modules/motifs, and amino-acid runs or ‘homopeptides’. Such residue patterning in IDRs can have functional significance, e.g., repeats in the Sup35p prion determinant linked to chaperone-dependent prion maintenance, but not to prion nucleation or fibre growth [14]; patterning of charged residues into blocks (i.e., residue ‘blockiness’) in transcriptional regulators and nucleolar proteins [15,16]; homopeptide content in transcriptional activators [17]; compositional modularity in stress-response proteins such as the water stress sensor protein FLOE-1 in *Arabidopsis* [18]. Some tools have been developed in recent years to tackle characterizing such patterning. The program NARDINI was developed to analyze specific types of binary compositional patterning in IDRs [19]. The CIDER package calculates the variation in sequence parameters such as hydrophobicity, net charge per residue, patterning of charged residues, and proline content for IDRs [20]. Particularly for CBRs, the program LCD-Composer can be applied to analyze both compositional bias and residue dispersion, the inverse concept to ‘blockiness’ [21], and the LCT server analyses ‘low complexity’ and distance to perfect repeat structure [22]. Blockiness and homopeptide content were demonstrated to have strong functional associations for intrinsically disordered CBRs in *Saccharomyces cerevisiae* [12].

To aid those venturing deeper into the dark proteomes, I have assembled a troupe of five decipherment helpers, collectively called ***Patterny***. These are programs that each focus on one particular feature of IDRs and CBRs, as listed above in the Abstract. They have been applied to three large data sets, and the penetrance of the analysed phenomena are discussed. Some illustrative examples are probed in more detail.

## 2. Materials and Methods

### 2.1. Data Sets

The DISPROT database of intrinsically disordered regions was downloaded in FASTA format in July 2025 [23]. This was reduced for sequence redundancy using an algorithm previously described [24], yielding a set of 6643 sequences. The proteome of budding yeast *Saccharomyces cerevisiae* (strain 288c) was obtained at the same time from UniProt (release 2025_3) [25]. For comparative purposes, the ASTRALSCOP40 data set of protein domain sequences (corresponding to version 2.08 of the SCOPe database) was also analyzed [26]. Orthologs of the illustrative example Chromogranin-A were taken from the OrthoDB database [27], and from previously calculated fungal ortholog sets by the author for the other example MSA2 [28].

### 2.2. Annotation of Compositional Bias Using fLPS2

The fLPS2 algorithm was used to label compositional biases and low-complexity regions [29,30]. It uses three main parameters, a minimum window size (**m**), a maximum window size (**M**) and a binomial *p*-value threshold (**t**), and applies a process of binomial probability minimization. At the end of the process, single- and multiple-residue compositional biases are output if they are below the user-specified *p*-value threshold, or default threshold. Biased regions are labelled with a *bias signature* which is a list of the biasing residues in order of bias precedence delimited with curly brackets. At various algorithmic stages, efficiency measures are taken to avoid or delay probability calculations unless/until they are necessary. The program fLPS2 was updated to include a FASTA format output (-f option), and an option (-b) to allow minimum window sizes down to 3, which is used for analysis of banding (see below).

### 2.3. Patterny Flow Design

The ***Patterny*** flow is drawn in Figure 1A. Submitted sequence data is assessed by each program individually, however output of the *Moduley* program is further fed into the *Blocky*, *Repeaty* and *Runny* programs. *Bandy* operates separately. The individual programs/scripts are described below.

### 2.4. Moduley: Labelling Compositional Modules and Other Possible Compositional Boundaries

Compositional modules (CModules) are defined as regions of compositional bias, optimized over a range of possible parameter sets. *Moduley* performs this definition task (Figure 1B). For this, a list of twelve fLPS2 parameter sets (**m**, **M** and **t** values) that were applied to thoroughly picking apart the functional associations of intrinsically disordered compositional biases (Appendix A), were re-applied here. These parameter triads cover a range of target lengths and estimated data set coverages for compositionally biased regions [11,12]. All the annotated compositionally biased regions from all the outputs are sorted on increasing *p*-value. Then for any one region, any other region with the same primary bias (i.e., most dominant residue type) and overlap over most of its extent (≥0.5) is de-selected. This progressive de-selection continues until there are no more regions to assess.

In parallel, larger lists of *boundary sets* are formed through an analogous de-selection procedure, except the criterion for overlap is to have both ends within a small margin (=5 residues was found to be suitable).

There is one flag for the ***Patterny*** script (‘–CModules yes|no’), which can be used to turn off calculation and analysis of compositional modules, e.g., if a previously calculated set of them is being digested.

### 2.5. Bandy: Discerning Compositional Banding

Compositional banding occurs when two or more patches of the same primary compositional bias are detected in an input sequence. Bandy has been designed to pick out sets of bands and to assess how evenly arranged these bands are. To discern band sets, a new option in the fLPS2 program was applied (‘-b’) which allows for minimum window lengths down to 3, while keeping maximum window sizes ≤ 20. A set of twelve parameter sets using very smaller window sizes was applied, and the resulting annotations were pooled and then segregated according to their primary residue bias, or both primary and secondary residues biases for multiple-residue biased regions (Figure 1C).

Each band set was then assessed for its *distance to perfect banding* (*DPB*). This is calculated by: (1) re-distributing the endpoints of the bands evenly over the same overall span; (2) calculating the deviation of each original endpoint to its corresponding ‘perfect’ endpoint; (3) summing these deviations to get DPB. The original DPB values are then compared to the DPB values arising from a sample of 1000 random endpoint sets of the same number placed along the same span, to derive z-scores and *p*-values (Figure 1C). In doing so, for band sets with band number ≥ 4, outlier intervals between bands are labelled and excised if their median absolute deviation is ≥3.5. Finally, for each primary bias, the following are output: (a) the band set with the highest band number (if there is a tie, the one with the smallest *p*-value is picked); (b) the band set with the lowest z-score; (c) the band set with the highest z-score.

### 2.6. Blocky: Assessing Residue Segregation

The distribution of residues along the expanse of an IDR or CBR can vary quite substantially. One aspect is their degree of bunching or ‘blockiness’. At one extreme, the most ‘blocky’, all the residues are segregated from each other in decreasing order of frequency from one end of the sequence to another. At intermediate levels of blockiness, there may be smaller residue ‘islands’; whereas at the other extreme, amino acids of a specific type try to be as distant as possible from their fellows. The *Blocky* algorithm to calculate blockiness was described previously [12]. It calculates a blockiness score (***B***), which is an indicator of how segregated residue types are along an input sequence (Figure 2A). The formula for this is:$$B=\frac{\sum_{i=1}^{L}d_{min}^{diff}}{\sum_{i=1}^{L}d_{min}^{same}}$$
where **L** is the length of the sequence being considered, and $d_{min}^{diff}$ is the smallest interval from residue **i** to any residue of a different type, and $d_{min}^{same}$ is the smallest interval to a residue of the same type.

Originally, it was normalized using time-consuming calculations of minimum possible blockiness. Here, it is simplified relative to its previous treatment, so that only the raw score **B** is considered, but also now it is compared against values calculated for 1000 scrambled sequences of the same composition. From the ***B*** distribution, z-scores and *p*-values are calculated. Where residue-specific **B** values are >the overall **B** value, this indicates that the residue is contributing to the residue segregation tendency.

### 2.7. Runny: Measuring Homopeptide Content

Homopeptides are defined as runs of amino acids of the same type with a minimum length of 3 residues [31]. Previously we dissected the intimate connection between homopeptide content (abbreviated **hpep** in the program outputs) and the function of intrinsically disordered compositionally biased regions (ID-CBRs) [12]. Here, Runny calculates homopeptide content and assesses its significance relative to a population of 1000 scrambled sequences of the same composition, as above for Blocky (Figure 2B).

### 2.8. Repeaty: Calculating Repetitiveness

Repeaty calculates the overall repetitiveness of a sequence using a concept of residue interval entropy (***IE***) drawn up here, which is given by:$$IE=\sum_{i\leq N}(-p_{i}.\log_{2}p_{i})$$
where there are ***N*** types of residue interval. N comprises all possible interval types of the sort x…[δ]…z, where the interval δ is in the range 0 to 100, and the residue pairs x and z are all possible pairings, including those with x = z. To make the calculation computationally tractable, only intervals between residue types that occur at least three times in the sequence are considered. As above for Runny and Blocky, the significance of the value of **IE** is assessed relative to a population of 1000 scrambled sequences of the same composition (Figure 2C). **IE** values just for intervals with x = z (same-residue) and x ≠ z (different-residue) are also determined.

In addition to these overall **IE** values, an ‘experimental’ output of the top ten intervals contributing most to **IE** is provided, sorted in two different ways: (1) any significant interval, but sorted on decreasing frequency; (2) sorted on significant *p*-value.

### 2.9. The Patterny Script and the Program Implementations

Each of the components of **Patterny** are written in C and shell script (with one short AWK script), and executed using a shell script (patterny, either BASH or zsh, there are no shell-specific commands). The current version of fLPS2 (2.1, described above) has also been updated. The package is available from Github [https://github.com/pmharrison/patterny/], and includes some examples input and output files. The details of program execution and output format appear in the README.

## 3. Results & Discussion

### 3.1. Rationale, Test Data Sets & Performance

*Patterny* is a troupe of decipherment helpers designed to provide information which may guide further inquiry and hypothesis generation for protein regions whose function is encoded in a distributed manner, such as IDRs, and LCRs/CBRs more generally. Currently, there are five members in the troupe that focus on different distributed properties. Firstly, *Moduley* discovers optimized compositional modules, and also longer lists of *Boundary Sets* for compositionally defined regions. The latter may be useful for picking more sensible tracts to piece together for experimental constructs, or someone might even be keen on applying them to more thorough bioinformatical analyses. Secondly, *Bandy* labels *compositional banding*, which occurs when there are at least two tracts with the same primary amino-acid compositional bias. Thirdly, *Blocky* assesses the overall segregation of residues by type along a sequence tract (blockiness). Fourthly, *Runny* highlights sequence tracts that have significant enrichments (or occasionally, lacks) of homopeptides, i.e., runs of amino-acid residues ≥ 3 in size [31]. Both the latter properties were demonstrated to have clear functional associations for tracts with the same primary bias in the model organism *Saccharomyces cerevisiae* [12]. Fifthly, *Repeaty* measures the overall repetitiveness of a tract using a novel conception of residue interval entropy, and provides output that highlights the most prominent residue intervals. *Repeaty* assesses repetitiveness without explicitly pulling predicted repeats out of the input.

Two data sets were derived for testing *Patterny*: (1) the DISPROT database of intrinsically disordered regions found by experiment was reduced using a clustering procedure previously developed [24], to make it non-redundant (DISPROT_NR_); (2) a set of compositional modules from the *S. cerevisiae* (budding yeast) proteome found by the *Moduley* program (CModules_YEAST_). A third set of structural protein domain sequences, ASTRALSCOP40 takes on the role of a comparative ‘control’.

The performance of the package was checked for the DISPROT_NR_ set (6643 sequence tracts) and the CModules_YEAST_ set (24,043 sequence tracts). The full ***Patterny*** package takes 39.1 s system time to process DISPROT_NR_ and 103.5 s for the CModules_YEAST_, with ~>90% of these timings being taken up with the *Repeaty* program. For CModules_YEAST_, derivation and assessment of compositional modules is not carried out (‘–CModules no’). These timings were assessed on a 2020 Apple Mac Mini with an M1 chip and 16 GB RAM. The package can thus analyze large databases and proteomes quite tractably.

### 3.2. Prevalences of Features in the DISPROT_NR_ Set

To gauge the penetrance of the phenomena explored, I summarized all the results for the three data sets in a big table (i.e., Table 1). CModules are a common feature of every data set, but the average −log(*p*-value) of the compositional bias is substantially lower for the larger abundance of them in the ASTRALSCOP40 structural domain sequences (~4.4, i.e., *p*-values of about 10^−4^), compared to DISPROT_NR_ (~8.3), and CModules_YEAST_ (~6.4). Compositional banding occurs for about 1 in 10 of sequences regardless of origin, and there is even a handful of banding patterns ≥ 3 in number and ‘significantly uneven’ (0.2% in DISPROT_NR_ and CModules_YEAST_). Significant blockiness and homopeptide content are most common in DISPROT_NR,_ with significant repetitiveness actually most common in ASTRALSCOP40, but ≥9% frequency for all three data sets. Again, there are diminutive handfuls of sequences that are significantly un-blocky, un-repetitive or lacking in homopeptide content (Table 1).

Since >11% of ASTRALSCOP40 domains have significant repetitiveness, a list of the protein folds that are most often repetitive was compiled (Appendix A). This list is topped by the leucine-rich repeat domain, and includes several other domains with internal structural repeats, such as beta-propellers and the triose phosphate isomerase beta/alpha-barrel.

Are *Blocky* and *Bandy* redundant in utility? Maximum blockiness occurs when residues are perfectly segregated by type. Perfect banding occurs when a residue type occurs in bias bands that are perfectly spaced. However, only about 21% (90/430) of the DISPROT entries that have ‘significantly even’ bands are also ‘significantly blocky’ by the *Blocky* algorithm, indicating some overlap, but a substantial difference in emphasis.

To check whether it is sufficient to generate 1000 scrambled sequences for these calculations, a 10% sample of the DISPROT_NR_ data set was fed into *Patterny* 100 times over. For each calculation for these runs, the variance in percentages reported in Table 1 is no more than ±0.3%. Nonetheless, if a user would like greater accuracy or fears marginal results, the scrambled sequence generation can be increased by adjusting the SAMPLE_SIZE variable within each program in the package.

### 3.3. Ranges of Behaviour for the Properties Explored

Some ‘toy’ sequences with obvious properties were submitted to ***Patterny*** for illustrative purposes (Figure 3). These four toy sequences have obvious tandem repeats, blocks of residues, alternating bands, and the fourth is a randomly generated sequence (each of five residue types being equally likely). The sequence with alternate bands (sequence (C)), has bands that are evenly spaced, and has according to *Blocky* much less extreme blockiness than the one with discrete blocks (sequence (A)), and also registers as repetitive according to the interval entropy calculation by *Repeaty*, solely because of intervals between residues of different types. Sequence (A), while extremely repetitive, has a much weaker ‘un-blocky’ signal detected by *Blocky*; thus *Repeaty* captures an essential property of this sequence much more clearly.

As a validatory exercise, extremes of modularity, blockiness, band evenness, homopeptide content and repetitiveness were examined. These are listed in Appendix A. The output from each example for the relevant ***Patterny*** program has been isolated along with its sequence (the description of the headers is given in detail in the README bundled with the package). Just to highlight a few of these examples, firstly, an extreme case of modularity is the frequency clock protein from *Neurospora crassa*, which has 16 compositional modules which do not merge into a larger module, such as is observed in the C-terminal fragment of S/A-repeat-containing protein D from *Staphylococcus aureus* [33,34]. Disprot entry DP01621r005 (the C-terminal IDR of the LANA protein from Herpesvirus 8) is both extremely un-blocky and devoid of homopeptides [23]. To demonstrate the effectiveness of IE at ascertaining repetitiveness, the most extreme value is observed for the central disordered fragment from Nucleoporin NSP1 from S. cerevisiae (DP01077r015), with a z-score of −46.1) [35]. This protein contains several large exact repeats.

### 3.4. Detailed Example from DISPROT: Chromogranin-A from Domestic Cod

Two detailed random examples of no particular interest were picked from each data source. The output files for these examples are available at Github [https://github.com/pmharrison/patterny/tree/main/Examples/output, accessed on 15 September 2025]. Firstly, the protein chromogranin-A from *Bos taurus* (domestic cow) (entry DP00118r011 from DISPROT). Chromogranin A is a multi-functional precursor that, through its proteolytic cleavage, generates a family of biologically active peptides that collectively exert regulatory effects on diverse physiological systems in vertebrates. It is experimentally demonstrated to be ~100% intrinsically disordered [23,36]. It was probed for a small panel of vertebrates. The compositional modules observed in it are drawn in Figure 4A, with other ***Patterny*** outputs summarized in 3B. Significant blockiness is a feature of chromogranin-A for several vertebrates, and any blockiness observed is centred around E, L, A and S residues chiefly.

There are notable homopeptide enrichments in the mammalian sequences generally for E-homopeptides, and for Q-homopeptides specifically only in mice. The sequences have a significant conserved repetitiveness across all species that stems largely from intervals between different residue types. These results are observed for both the whole sequences and the largest compositional modules within each sequence.

### 3.5. Detailed Example from CModules_YEAST_: Putative Transcriptional Activator MSA2 from S. cerevisiae

The second example is MSA2 a putative transcriptional activator that along with its paralog MSA1 is a key regulator of the G1/S transition of the cell cycle. MSA2 originated after the whole-genome duplication of budding yeasts; it is sporadically conserved across *Saccharomycetaceae*, and has originated since the last common ancestor of that clade. Alphafold predicts it as almost completely disordered, with intermittent alpha helices (for reference, please see its UniProt database record, accession P36157 [25]). Here, we observe that MSA2 has a core compositional module that tends to contain S, N and P residues (Figure 5A). There is compositional banding for N residues across most species, and conserved significant blockiness is observed in *Saccharomyces* (S_*) and *Naumovozyma* (N_*) species that is mainly caused by segregation of S, N, T, P and K residues, while significant homopeptide content **(hpep)** is observed in most of the species, most notably for S and N residues. Unlike the DISPROT example, there is not a conserved significant repetitiveness over the whole data set (just within the Saccharomyces genus, and in N. glabratus), either across whole orthologs or just within compositional modules.

In a recent analysis of ID-CBRs (intrinsically disordered compositionally biased regions), the short Q-rich tract in MSA2 is linked to clusters with a possible function in regulation of transcription by RNA polymerase II (GO:0006357), and the N-rich tract to various categories linked more generally, or more specifically to regulation of transcription (e.g., GO:0006355, GO:0045944, GO:0001228) [12,37].

### 3.6. More Examples (With Experimentally Characterized Compositional Tracts)

I picked three more examples that have experimentally characterized tracts, to demonstrate how the compositional modules calculated here correspond to them, and to discuss the properties that **Patterny** has pulled out of them. The Uniprot accessions for each protein are provided, so that the reader can regenerate the results. Firstly, the Sup35p protein from S. cerevisiae that underlies the [PSI+] prion and functions in translation termination was examined (Uniprot accession P05453) [38,39]. In Sup35p, Moduley identifies a QYNG-rich compositional module corresponding to the experimental prion determinant and an EK-rich module that corresponds to the M-domain that functions in biomolecular condensation [40,41]. Intriguingly, the QYNG-rich compositional module is not significantly repetitive relative to the randomly scrambled sequences, but does have two N-rich bands evenly placed at either end (residues 5–21 and 93–109). Four other short compositional modules are also identified, with two of the them linked to homopeptides.

Secondly, I examined the compositional modules in the FLOE1 protein (Uniprot accession Q8VZR8), a prion-like protein regulator of seed germination that undergoes hydration-dependent phase separation [18]. Distinct experimentally relevant tracts in FLOE1 are identified by Moduley, namely a long C-terminal PQ-rich module linked to condensate formation, and a SD-rich N-terminal module linked to gain of function behaviours, along with four short modules ≤ 20 residues long. There is no significant repetitiveness, blockiness, homopeptide content, etc., within these modules, except for a serine homopeptide in one short one.

Thirdly, human collagen alpha-1(VII) chain was probed for compositional modules (Uniprot accession Q02388). The triple-helical region that functions as a structural scaffold is labelled as a highly repetitive and GP-rich compositional module, while two other compositional modules correspond to the fibronectin-domain region that facilitates cell-to-collagen interaction, the longest of which also demonstrates significant repetitiveness. Five other short compositional modules (≤25 residues) are also picked out, three of which are linked to homopeptide content.

### 3.7. Further Examples

Some further examples (four each from DISPROT_NR_ and CModules_YEAST_) and their outputs are bundled with the package, and available at Github [https://github.com/pmharrison/patterny/tree/main/Examples]. These demonstrate diverse traits in terms of modularity, banding, blockiness, homopeptide content and repetitiveness. For example, there is an EK-rich compositional module (that was also identified as an intrinsically disordered CBR in ref. [12]) in mannosyltransferase regulator 4, which operates in N-glycan mannosylphosphorylation (a functionality only found in fungi), that has obvious E and K banding and high homopeptide contents.

### 3.8. Patterny Source Code Distribution

The **Patterny** source code, some executables and the example data is available at Github [https://github.com/pmharrison/patterny/]. The details of output formats can be found in the README.txt bundled with the package.

## 4. Conclusions

It is hoped that this package might be useful for hypothesis generation for IDRs and CBRs in proteins. The ***Patterny*** outputs could be used to guide mutations and molecular constructs in laboratory experiments. Indeed, in addition to the short-listed compositional modules, there are longer lists of possible compositional ‘boundary sets’ that might be useful for specifying boundaries for constructs. Also, computational biologists could graft the package into pipelines to probe large-scale data sets, such as proteomes, for the functional manifestations of IDR and CBR features using the compositional module paradigm. Such pipelines could also include inputs from intrinsic disorder annotation tools. However, for such analysis, CBRs still need to be differentiated according to bias and length, or according to the program parameters used to annotate them [12]. ***Patterny*** is complementary to tools such as CIDER or NARDINI which specifically focus on the calculation of compositionally biased patches and patterns within IDRs [19,20], but has the added advantage of being unreliant on specific window sizes, and it also can automatically define and characterize compositionally biased modules of any type, whether intrinsically disordered or not. Results are presented for both compositional modules and whole input sequences, which increases the utility of the ***Patterny*** package for experimental hypothesis generation.

Several further developments of the package are anticipated. Firstly, the sort of linear regression that was used in a previous study of yeast intrinsically disordered CBRs [12] will be implemented more generally. Also, the package will gain further power through the lens of phylogeny trees, and explicit consideration of clade-specific conservation of traits. Such phylo-optical intensification of the algorithms will hopefully yield insights when cross-referenced with functional information, e.g., from Gene Ontology [37].

## Figures and Tables

**Figure 1 biomolecules-15-01332-f001:**
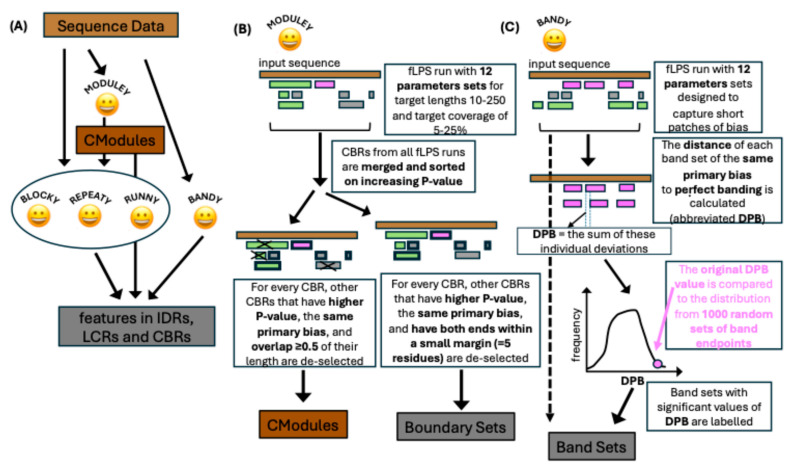
Patterny flow and the *Moduley* and *Bandy* algorithms. (**A**) Overall flow. Input sequence data is submitted to *Moduley* to make sets of CModules and a larger list of boundary sets. In parallel, *Bandy* speculates about bands of similar composition in each sequence of the input data. *Blocky*, *Repeaty* and *Runny* process both the original input data, and the sets of CModules from *Moduley*. (**B**) A graphical depiction of the *Moduley* algorithm described in Section 2.4. (**C**) A graphical depiction of the *Bandy* algorithm described in Section 2.5.

**Figure 2 biomolecules-15-01332-f002:**
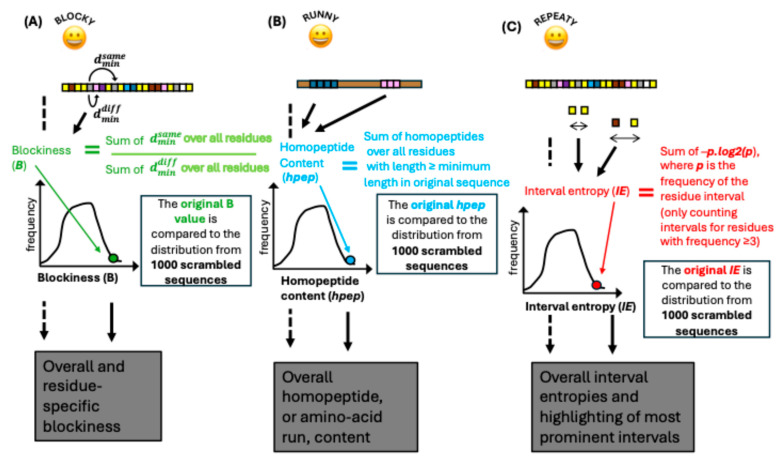
Graphical depictions of the algorithms of (**A**) *Blocky*, (**B**) *Runny* and (**C**) *Repeaty*. These are described in detail in Section 2.

**Figure 3 biomolecules-15-01332-f003:**
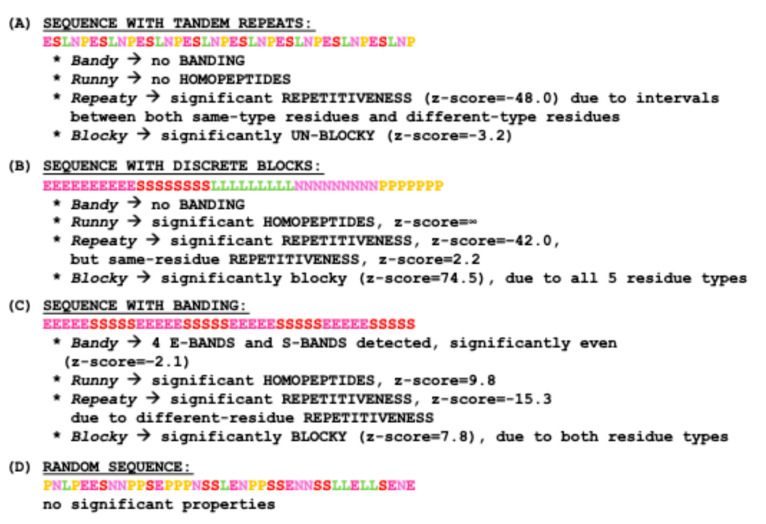
The properties of four toy sequences submitted to *Patterny*: the sequences are coloured using the Taylor amino-acid residue scheme [32]. They are as follows: (**A**) a sequence with tandem repeats, (**B**) a sequence with discrete residue blocks, (**C**) a sequence with alternate bands, and (**D**) a randomly generated sequence.

**Figure 4 biomolecules-15-01332-f004:**
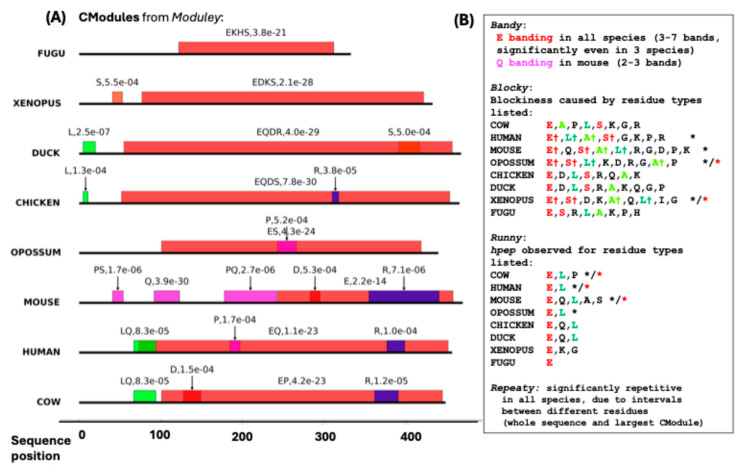
Example from the DISPROT database: Chromogranin-A from *B. taurus* (cow). (**A**) A picture of the compositional modules in each of the eight vertebrate orthologs analyzed. These are coloured using the Taylor amino-acid residue scheme [32], and labelled with their bias signatures and bias *p*-values. (**B**) A summary of the output for the other four programs. Significant blockiness or homopeptide content relative to the sample of 1000 scrambled sequences is indicated with a black asterisk (*), and with a red asterisk if they are also significant in this way for the largest compositional module in each sequence. The symbol † labels residues that contribute to blockiness and occur across all those labelled *.

**Figure 5 biomolecules-15-01332-f005:**
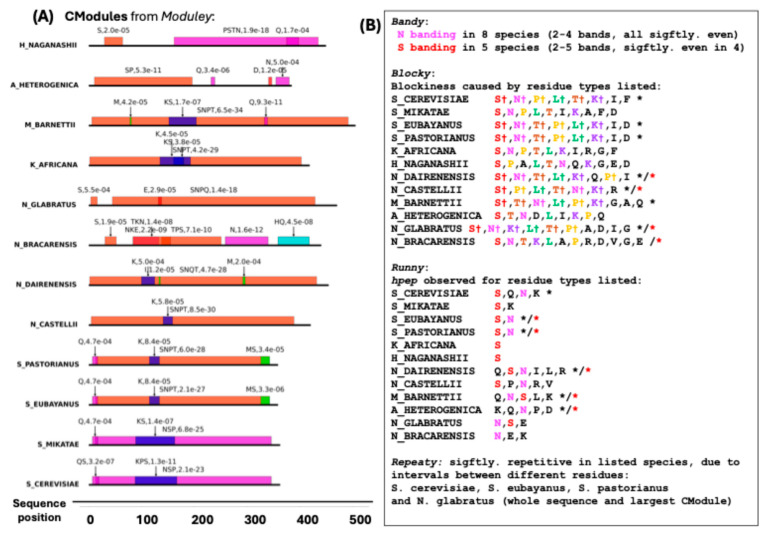
Example from the CModules_YEAST_ data set: MSA2 putative transcriptional activator from *S. cerevisiae*. (**A**) A picture of the CModules in each of the twelve *Saccharomycetaceae* orthologs analyzed. These are coloured and labelled as in Figure 4. (**B**) A summary of the output for the other four programs. Significant blockiness or homopeptide content relative to the sample of 1000 scrambled sequences is indicated with a black asterisk (*), and with a red asterisk if they are also significant in this way for the largest compositional module in each sequence. The symbol † labels residues that contribute to blockiness and occur across all those labelled *.

**Table 1 biomolecules-15-01332-t001:** Prevalences.

Feature	Data Sets → DISPROT_NR_ (Total = 6463)	CModules_YEAST_ (Total = 24,043)	ASTRALSCOP40 (Total = 14,844)
Modularity: ** CModules (≥1) CModules (≥2) CModules (≥3)	2454 (38.0%) 746 (11.5%) 309 (4.8%)	--- --- ---	**10,096 (68.0%) *** **4490 (30.2%)** **1794 (12.1%)**


Banding: Bands (≥2) Bands (≥3) Significantly even bands (≥2) Significantly uneven bands (≥2) Significantly even bands (≥3) Significantly uneven bands (≥3)	**670 (10.4%)** **389 (6.0%)** 430 (6.7%) 17 (0.3%) 66 (1.0%) 14 (0.2%)	**3217 (13.4%)** **1640 (11.0%)** 2145 (8.9%) 59 (0.2%) 244 (1.0%) 51 (0.2%)	1317 (8.9%) 382 (2.6%) 950 (6.4%) 1 (0.0%) 46 (0.3%) 1 (0.0%)

Blockiness (***B***): Significantly blocky Significantly un-blocky			
	**560 (8.7%)**	**1592 (6.6%)**	575 (3.9%)
	70 (1.1%)	51 (0.2%)	38 (0.3%)

Homopeptide content (***hpep***): Significant enrichment Significant lack	**802 (12.4%)** 9 (0.1%)	**2280 (9.5%)** 16 (0.1%)	938 (6.3%) 0 (0.0%)
Repetitiveness (**IE**): Significantly repetitive Significantly un-repetitive	598 (9.3%) 54 (0.8%)	2194 (9.1%) 269 (1.1%)	**1672 (11.3%)** 136 (0.9%)


* Entries are in bold if either the ASTRALSCOP40 set or both the DISPROT_NR_ and CModules_YEAST_ set have higher occurrences. ** The properties examined are underlined.

## Data Availability

Data used as examples were downloaded from the UniProt, DISPROT and OrthoDB databases.

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
