# Peer review of "Patterny: A Troupe of Decipherment Helpers for Intrinsic Disorder, Low Complexity and Compositional Bias in Proteins"

_biomolecules, 2025, doi:10.3390/biom15091332_

Round 1

Reviewer 1 Report

Comments and Suggestions for Authors

In the manuscript titled “Patterny: A troupe of decipherment helpers for intrinsic disorder, low complexity and compositional bias in proteins”, the author presents a carefully assembled troupe of algorithms designed to unravel the intricacies of intrinsic disorder, low-complexity regions, and compositional bias within protein sequences. This collective of decipherment tools holds promise for advancing the identification and characterization of IDPs, IDRs, and CBRs. In this context, I would like to raise a few suggestions in the form of questions, which, if addressed, could further enrich the manuscript and broaden its resonance with the scientific community.

  1. The first question concerns the distinction of Patterny from pre-existing algorithms such as PONDR and CIDER. How does Patterny rise above these established methods—whether in efficiency, accuracy, versatility, or interpretability? Clarifying its unique strengths would not only underscore its novelty but also highlight the added value it brings to the field.

  2. A second point relates to the references: of the 29 citations, 7 are self-citations (nearly 24%). While this may reflect continuity of the author’s own contributions, it also suggests a certain insularity. Moreover, the very design of Patterny as a troupe—a collection where individual modules like Cmodule can stand alone—may inadvertently create a sense of complexity for prospective users. Could such modularity, though powerful, risk deterring a broader audience who seek accessibility and streamlined usability? Addressing how the package balances its richness of tools with clarity, user-friendliness, and integration would greatly enhance its appeal.

Author Response

In the manuscript titled “Patterny: A troupe of decipherment helpers for intrinsic disorder, low complexity and compositional bias in proteins”, the author presents a carefully assembled troupe of algorithms designed to unravel the intricacies of intrinsic disorder, low-complexity regions, and compositional bias within protein sequences. This collective of decipherment tools holds promise for advancing the identification and characterization of IDPs, IDRs, and CBRs. In this context, I would like to raise a few suggestions in the form of questions, which, if addressed, could further enrich the manuscript and broaden its resonance with the scientific community.

  1. The first question concerns the distinction of Patternyfrom pre-existing algorithms such as PONDR and CIDER. How does Patterny rise above these established methods—whether in efficiency, accuracy, versatility, or interpretability? Clarifying its unique strengths would not only underscore its novelty but also highlight the added value it brings to the field.

Response: The emphasis of Patterny is distinct from PONDR and similar programs. Programs like PONDR are for the prediction of intrinsic disorder, whereas Patterny is for the characterization of patterns in intrinsically-disordered regions (IDRs) and compositionally-biased regions generally. I have added some text and references for this activity in the Introduction.    The programs CIDER and NARDINI are specifically for the characterization of compositionally-biased patches and patterning of combinations of residues in IDRs. These tools require the choice of specific window sizes for the analysis, whereas Patterny does not. Patterny is also more comprehensive and versatile in that it can define and characterize any sort of compositionally-biased module automatically (whether in IDRs or not). I have added some text in ‘Conclusions’ about this. The question of ‘accuracy’ is not relevant for this sort of analysis since there are no ‘positives’ and ‘negatives’; the results are simply to guide hypothesis generation.

  1. A second point relates to the references: of the 29 citations, 7 are self-citations (nearly 24%). While this may reflect continuity of the author’s own contributions, it also suggests a certain insularity. Moreover, the very design of Patternyas a troupe—a collection where individual modules like Cmodule can stand alone—may inadvertently create a sense of complexity for prospective users. Could such modularity, though powerful, risk deterring a broader audience who seek accessibility and streamlined usability? Addressing how the package balances its richness of tools with clarity, user-friendliness, and integration would greatly enhance its appeal.

Response: The ‘self-citations’ are to papers that describe algorithms or data that are used in the paper. More citations to other sorts of computational analysis of IDRs have been added, as suggested (see above). I have added examples illustrating how the compositional modules defined here can correspond to experimentally characterized functional tracts (section 3.6). The ‘Conclusions’ section has been expanded to address utility and versatility.

Reviewer 2 Report

Comments and Suggestions for Authors

This manuscript reports on a suite of software tools used to examine
properties of protein sequences, geared toward identifying
characteristics of intrinsically disordered, or low-complexity
regions. This seems to be a potentially useful contribution and is
well-written.

The manuscript describes a snapshot of a long-running campaign of
developing software tools for sequence analysis. As such, previous
publications must be consulted in order to understand the
algorithms. A bit deeper summary of these algorithms would make for
smoother comprehension of this work. 

I have verified that the software is available on github, is
downloadable and runnable.

I have the following suggestions:

- Table 1 should corrected so that the lines align.

- The text in all of the figures is too small to read in single page
  letter or A4 format.

- I suggest that the README.txt (on github) be modified to specify a
  run command of ``bash'' instead of ``sh''.

Author Response

This manuscript reports on a suite of software tools used to examine
properties of protein sequences, geared toward identifying
characteristics of intrinsically disordered, or low-complexity
regions. This seems to be a potentially useful contribution and is
well-written.

The manuscript describes a snapshot of a long-running campaign of
developing software tools for sequence analysis. As such, previous
publications must be consulted in order to understand the
algorithms. A bit deeper summary of these algorithms would make for
smoother comprehension of this work. 

Response: I have added some more detail about the fLPS and Blocky algorithms in Materials & Methods.

I have verified that the software is available on github, is
downloadable and runnable.

I have the following suggestions:

- Table 1 should corrected so that the lines align.

Response: The table alignment has been corrected.

- The text in all of the figures is too small to read in single page
  letter or A4 format.

Response: Text in the figures has been enlarged.

- I suggest that the README.txt (on github) be modified to specify a
  run command of ``bash'' instead of ``sh''.

Response: This has been changed as suggested.

Reviewer 3 Report

Comments and Suggestions for Authors

The manuscript describes the development of a software package for the analysis of low complexity regions in proteins. The exact analysis of such segments is important and the tools described represent a useful contribution to the field.

I have some suggestions to improve the presentation of the programs in the manuscript.
- fLPS2 and Blocky have previously described earlier versions. I suggest adding a brief summary of the principles behind fLPS2 and even including the twelve parameters in the present manuscript (either in the main text or as a supplementary material) instead of just providing references. For Blocky, I suggest including the definitive formula of B in the main text also.
- In general, I suggest adding simple sequence-based examples (not necessarily from actual proteins) do demonstrate and explain the algorithms. In this respect, the relationships between the boundaries, repetitiveness and blocks could be shown for the prospective users on sample (e.g. colored) sequences.
In this respect, the simple txt file provided as supplementary is not straightforward to interpret.
- I suggest that the author estimates the reproducibility of Z-scores calculated from the 1,000 randomizations – for distinct runs, each with 1,000 randomization, the Z-scores might slightly differ and it would be good to assess the expected upper limit of the variability.
- The author states that repetitivity is often observed in the sequences of the structured domains investigated. It would be nice to provide some examples and discuss this issue further, especially the relationships between the sequence features investigated by Patterny and globularity, and whether there are specific protein classes (like helical repeat proteins?) for which this repetitivity is characteristically high.
- I suggest enhancing the demonstrative analysis of the examples. It would be nice to see explicitly that the toolkit developed can identify regions that have experimentally described different functionality – this can be shown using selected well-known proteins maybe. It would also be nice to see how the tools perform on selected known repetitive motifs like coiled coils or polyproline repeats characteristic for collagens. I also suggest proposing an analysis pipeline in which Patterny is placed in context with other tools (such as disorder predictors) both as a practical guide and to emphasize the added value of Patterny further. 

Minor issues:
For better readability, I suggest writing out “homopeptide content” in most cases instead of “hpep”. The same might be considered for CModules, especially as it is written in bold like the program names and is a bit confusing. Writing “Compositional modules” would make the text better readable still without excessively long phrases.
Line 265: Please add the Uniprot ID for the MSA2 test case. 
Line 271: Please italicize Saccharomyces
Line 272: Please italicize N. glabratus

Author Response

The manuscript describes the development of a software package for the analysis of low complexity regions in proteins. The exact analysis of such segments is important and the tools described represent a useful contribution to the field.

I have some suggestions to improve the presentation of the programs in the manuscript.
- fLPS2 and Blocky have previously described earlier versions. I suggest adding a brief summary of the principles behind fLPS2 and even including the twelve parameters in the present manuscript (either in the main text or as a supplementary material) instead of just providing references. For Blocky, I suggest including the definitive formula of B in the main text also.

Response: Further details about the fLPS algorithm and the parameters have been added in Materials and Methods, including a supplementary table for the parameter triads. The Blocky formula for B has been added.

- In general, I suggest adding simple sequence-based examples (not necessarily from actual proteins) do demonstrate and explain the algorithms. In this respect, the relationships between the boundaries, repetitiveness and blocks could be shown for the prospective users on sample (e.g. colored) sequences.
In this respect, the simple txt file provided as supplementary is not straightforward to interpret.

Response: I added some ‘toy’ sequences with obvious properties (new Figure 3). There is some text discussing the results for these at the start of section 3.3.

- I suggest that the author estimates the reproducibility of Z-scores calculated from the 1,000 randomizations – for distinct runs, each with 1,000 randomization, the Z-scores might slightly differ and it would be good to assess the expected upper limit of the variability.

Response: This analysis has been added at the end of Section 3.2.  

- The author states that repetitivity is often observed in the sequences of the structured domains investigated. It would be nice to provide some examples and discuss this issue further, especially the relationships between the sequence features investigated by Patterny and globularity, and whether there are specific protein classes (like helical repeat proteins?) for which this repetitivity is characteristically high.

Response: A table of the most repetitive protein folds in the ASTRALSCOP40 domain data set was compiled (Suppl. Table 2), and is discussed in the text in section 3.2.

- I suggest enhancing the demonstrative analysis of the examples. It would be nice to see explicitly that the toolkit developed can identify regions that have experimentally described different functionality – this can be shown using selected well-known proteins maybe. It would also be nice to see how the tools perform on selected known repetitive motifs like coiled coils or polyproline repeats characteristic for collagens. I also suggest proposing an analysis pipeline in which Patterny is placed in context with other tools (such as disorder predictors) both as a practical guide and to emphasize the added value of Patterny further. 

Response: I have added a section entitled ‘More Examples’ (section 3.6) that discusses results for three proteins with experimentally-characterized tracts that correspond to compositional modules defined by Moduley. These are Sup35p from budding yeast, the FLOE1 protein from Arabidopsis and a human collagen molecule.

Minor issues:
For better readability, I suggest writing out “homopeptide content” in most cases instead of “hpep”. The same might be considered for CModules, especially as it is written in bold like the program names and is a bit confusing. Writing “Compositional modules” would make the text better readable still without excessively long phrases.

Response: A lot of references to hpep have been removed or replaced with ‘homopeptide content’ or some similar phrase. The bold type for CModules has been removed, and the term ‘compositional modules’ is used more often. The abbreviation CModules is used in a program flag and in outputs, so it is still mentioned where appropriate.

Line 265: Please add the Uniprot ID for the MSA2 test case. 
Line 271: Please italicize Saccharomyces
Line 272: Please italicize N. glabratus

Response: These have been amended.

Round 2

Reviewer 3 Report

Comments and Suggestions for Authors

I thank the author for his efforts in answering my points and significantly improving the manuscript with examples.